# Bacterial Colonization in Patients with Chronic Lymphocytic Leukemia and Factors Associated with Infections and Colonization

**DOI:** 10.3390/jcm8060861

**Published:** 2019-06-16

**Authors:** Izabela Korona-Glowniak, Ewelina Grywalska, Agnieszka Grzegorczyk, Jacek Roliński, Andrzej Glowniak, Anna Malm

**Affiliations:** 1Department of Pharmaceutical Microbiology, Medical University of Lublin, 20-093 Lublin, Poland; aga.grzegorczyk@umlub.pl (A.G.); anna.malm@umlub.pl (A.M.); 2Department of Clinical Immunology and Immunotherapy, Medical University of Lublin, 20-093 Lublin, Poland; ewelina.grywalska@gmail.com (E.G.); jacek.rolinski@gmail.com (J.R.); 3Department of Cardiology, Medical University of Lublin, 20-093 Lublin, Poland; andrzej.glowniak@gmail.com

**Keywords:** Gram-negative bacilli, *Staphylococcus aureus*, nasal and oropharyngeal colonization, respiratory infections, CLL patients

## Abstract

Patients with chronic lymphocytic leukemia (CLL) have defects in both humoral and cellular immunity as a result of their underlying malignancy, as well as chemotherapy-related immune suppression. Upper respiratory tract (URT) colonization can be regarded as a major contributor to infection, so the relationship between carriage rates, disease incidence, or antibiotic resistance should be monitored. This prospective study included 50 newly diagnosed, previously untreated patients with CLL and 38 healthy volunteers. A total of 264 samples obtained from anterior nares and oropharynx were microbiologically examined. A significantly higher frequency of *S. aureus* and Gram-negative bacilli (GNB) colonization in CLL patients was observed in comparison to healthy volunteers. Information regarding baseline characteristics; the Rai staging system; hematological tests results; immunophenotype of basic lymphocyte subsets, including the expression of programmed cell death-1 protein (PD-1) and its ligand (PD-L1); as well as Epstein-Barr virus (EBV) status were determined to analyze risk factors for infections and bacterial colonization. The data represent the basic information for identification of further risk factors of infection and bacterial oropharyngeal colonization in CLL patients. The rate of disease progression within the time from the CLL diagnosis was significantly higher in patients colonized by GNB. This study highlights EBV infection and frequencies of PD-1 positive T CD3^+^ cells and B cells as risk factors in CLL patients.

## 1. Introduction

Chronic lymphocytic leukemia (CLL) is the most common type of leukemia in European and North American adult populations which leads to significant immune system dysfunction [1,2]. Patients with CLL have defects in both humoral and cellular immunity as a result of their underlying malignancy, as well as chemotherapy-related immune suppression [3]. Hypogammaglobulinaemia is an important predisposing factor for infection in all CLL patients. The other factors include disturbances in both the specific (humoral and cellular) as well as innate immunity (dysfunction of the complement system, NK-cells, neutrophils and monocytes) [4]. The rate and risk factors for infections in CLL patients were determined in this study.

Upper respiratory tract colonization can be regarded as a major contributor to infection, so the relationship between carriage rates, disease incidence, or antibiotic resistance should be monitored. It is increasingly evident that the risk for colonization is the greatest in patients with serious underlying illness. Encapsulated bacteria (*Streptococcus pneumoniae*, *Haemophilus influenzae*) are the predominant pathogens in patients with CLL, but in the neutropenic post-chemotherapy phase also *Staphylococcus aureus* and various Gram-negative bacilli (GNB) such as *Pseudomonas aeruginosa, Escherichia coli* and *Klebsiella pneumoniae* can be responsible for bacteriema and septicemia, especially in patients with hypogammaglobulinemia [5]. A significantly higher frequency of GNB colonization in CLL patients was observed (36.7%) in comparison to healthy volunteers (8.3%) in our previous study [6]. The aim of this study was to identify the prevalence and the clinical and biochemical characteristics of CLL patients at higher risk of bacterial colonization including *S. aureus*, GNB, and upper respiratory pathogens (URT pathogens), namely *Haemophilus influenzae*, *Streptococcus pneumoniae*, and *Neisseria meningitidis*.

## 2. Experimental Section

### 2.1. Patients

This prospective study included 50 newly diagnosed, previously untreated patients with CLL and 38 healthy volunteers attending the Department of Clinical Immunology, Medical University of Lublin, from February 2015 to February 2016. The observation period amounted to 26 months (min. 12.50–max. 60), mean 28.25 ± 16.44. Information regarding baseline characteristics, stage according to the Rai staging system, hematological tests results were determined for CLL patients (Table 1) to analyze risk factors for infections and bacterial colonization. Neither the CLL patients nor the controls used immunomodulating agents or hormonal preparations, showed signs of infection within at least 3 months prior to the study, underwent blood transfusion, or presented with autoimmune condition or allergy. Moreover, none of the CLL patients and controls had a history of oncological therapy or prior treatment for tuberculosis or other chronic conditions that could be associated with impaired cellular or humoral immunity. All of the study participants were unvaccinated against *Streptococcus pneumoniae*, *Haemophilus influenzae*, *Neisseria meningitidis* or seasonal influenza. The diagnosis of CLL was established on the basis of diagnostic criteria included in the IWCLL guidelines of the American National Cancer Institute (NCI) [7,8]. The study was approved by the Ethics Committee of the Medical University of Lublin (decision no. KE-0254/227/2010). Written informed consent was obtained from all patients with respect to the use of their blood for scientific purposes.

### 2.2. Microbiological Procedures

Throat and nasal specimens were taken using sterile alginate-tipped swabs from each of the patients during control visit in clinical health care settings. Swabs were inoculated on Mueller-Hinton agar with 5% sheep blood, Mueller-Hinton agar with 5% sheep blood with 0.5 mg/L of gentamicin for selective cultivation of pneumococci, Haemophilus chocolate agar (BioMerieux, Marcy-l’Etoile, France) for selective cultivation of *Haemophilus* sp., Chapman agar for selective cultivation of staphylococci and McConkey agar, and selective medium for Gram-negative rods. Plates were incubated for 24–48 h at 35 °C under aerobic conditions. The streaked agar plates were incubated aerobically at 35 °C in 5% CO_2_ enriched atmosphere for 24 to 48 h. Pneumococci were identified by colony morphology, susceptibility to optochin (5 μg), and bile solubility; identification was confirmed by a slide agglutination test (Slidex Pneumo-Kit, BioMerieux, Marcy-l’Etoile, France). *N.meningitidis* and *H. influenzae* were identified by macroscopic, microscopic and biochemical assays by API NH microtest (BioMerieux, Marcy-l’Etoile, France ). Isolates of *S. aureus* were identified by colony morphology, biochemical activities (ID32 STAPH, BioMerieux, Marcy-l’Etoile, France), coagulase test, and a slide agglutination test (Slidex Staph-Kit, BioMerieux, Marcy-l’Etoile, France). The identification of Gram-negative rod isolates was determined using Api 20E or Api 20NE (BioMerieux, Marcy-l’Etoile, France) as appropriate.

### 2.3. Blood Sampling

Peripheral blood (15 mL) from the basilic vein of CLL patients and healthy controls was collected into EDTA-treated tubes (aspiration and vacuum systems Sarstedt, Numbrecht, Germany). Immediately after collection, the samples were used for immunophenotyping of lymphocytes, and for isolation of mononuclear cells for the EBV-DNA copy number determination. Isolation of mononuclear cells, isolation of DNA and determination of the EBV copy number, assessment of activated T and B cells, and analysis of CD38 and ZAP-70 expression in CLL cells were performed as described previously [9,10]. Viability below 95% disqualified the cells from further analyses.

### 2.4. Detection of Basic Peripheral Blood Lymphocyte Subsets

The percentages of cells expressing surface markers were analyzed as described previously [9,10]. Briefly, the cells were phenotypically characterized via incubation (20 min in the dark at room temperature) with a combination of relevant FITC–, PE–, and CyChrome–labeled mAbs. Immunofluorescence studies were performed using a combination of the following mAbs purchased from BD Biosciences (San Jose, CA, USA): CD45 FITC/CD14 PE, CyChrome Mouse Anti-Human CD3, FITC Mouse Anti-Human CD4, FITC Mouse Anti-Human CD8, FITC Mouse Anti-Human CD19, PE Mouse Anti-Human CD279 (PD-1), PE Mouse Anti-Human CD274 (PD-L1), PE-Cy5 Mouse Anti-Human CD25, and PE-Cy5 Mouse Anti-Human CD69. Three-color immunofluorescence analyses were performed using a FACSCalibur flow cytometer as described elsewhere [9,10].

### 2.5. Analysis of CD38 and ZAP-70 Expression in CLL Cells

CLL cells were stained for CD38 antigen and ZAP-70 protein expression (as described previously by Hus et al. [11]) and analyzed using flow cytometry. The cut-off point for CD38 and ZAP-70 positivity in leukemic cells was ≥30% and ≥20%, respectively.

### 2.6. DNA Isolation and Calculation of EBV Load

We isolated DNA from 5 × 10^6^ PBMCs with the QIAamp DNA Blood Mini Kit (QIAGEN, Hilden, Germany) according to the manufacturer’s instructions. The concentration and purity of the isolated DNA were verified with the BioSpec-nano spectrophotometer (Shimadzu, Kioto, Japan). The number of EBV-DNA copies in PBMCs was calculated with the ISEX variant of the EBV polymerase chain reaction (PCR) kit (GeneProof, Brno, Czech Republic). EBV DNA was analysed qualitatively and quantitatively with real-time PCR (RT-PCR). A specific conservative DNA sequence for the EBV nuclear antigen 1 (EBNA-1) gene was amplified with PCR. The number of viral DNA copies per μL of eluent was adjusted for the efficiency of DNA isolation, and then it was expressed as the viral DNA copy number per μg of DNA. All samples were examined in duplicate. A sample of pure buffer used for DNA elution was used as a negative control in every case. Because we used a method with a detection threshold of 10 EBV DNA copies per μl, all samples below this threshold were considered EBV-negative [EBV(–)]. PCR was performed with the 7300 Real Time PCR System (Applied Biosystems). The reaction was conducted on MicroAmp® Optical 96-Well Reaction Plates (Life Technologies) with MicroAmp® Optical Adhesive Film (Life Technologies).

### 2.7. Statistical Analysis

Data processing and analysis were performed using STATISTICA 10 (StatSoft. Inc.). The results are expressed as percentage or mean ± SD. Continuous variables were compared using a nonparametric test (*U*Mann-Whitney test) or parametric Student *t*-test and categorical variables by Chi-square or the Fisher exact test, as appropriate. Odds ratio (OR) and their 95% confidence intervals (CI) were calculated. The survival curves were constructed with the Kaplan-Meier method, and the proportions of survivors within the studied groups were compared with the log-rank test. Logistic regression models were fitted to identify risk factors associated with bacterial colonization. From these models, adjusted odds ratios (OR) and 95% confidence Intervals were derived; corresponding *p*-values were those from Wald’s test. Goodness of fit was checked using Hosmer and Lemeshow’s test. Statistical significance was set at *p* < 0.05. 

## 3. Results

Patients’ characteristics are given in Table 1. Among the types of infections, the most frequent both in the CLL patients and in the control group were upper respiratory tract infections (URTIs). Statistically significant higher numbers of infections and antibiotic therapies per year were observed in CLL patients comparing to healthy individuals (*p* < 0.0001). Noticeably, in CLL patients, lower respiratory tract infections (LRTIs) were more frequent than in the control group (*p* < 0.0001). Urinary tract infections (UTIs) and skin infections were observed in CLL patients only. The frequency and types of infections documented in CLL patients as well as clinical and laboratory parameters in detail are listed in Table 2.

Significantly higher risk of LRTIs in CLL patients was demonstrated for the following variables (Table 3): patients with Binet stages B (*p* = 0.0029); patients with the Rai stages 1 or 2 (*p* = 0.022 or 0.007, respectively); patients with hepatomegaly (*p* = 0.034); patients with high levels of β-2-microglobulin (*p* = 0.001) and lactate dehydrogenase (*p* = 0.04); patients with implemented treatment (*p* < 0.0001); patients with documented duplication of lymphocytosis (*p* = 0.002); patients EBV(+) meaning presented with a large number of EBV-DNA copies (more than 10 copies/uL) (*p* = 0.0006); patients with higher frequency of CD19+CD38 cells (*p* = 0.012) and patients with higher frequencies of CD4^+^/PD-1 within CD4^+^ (*p* = 0.001), CD8^+^/PD-1within CD8^+^ (p=0.03) and CD19^+^/PD-1 within CD19^+^ (*p* = 0.001). In the multivariable logistic regression model, only EBV(+) patients remained significant for LRTIs (OR 8.4, 95%CI 1.5–47.6, *p* = 0.016). 

A total of 264 samples obtained from anterior nares and oropharynx of 88 persons (50 patients with CLL and 38 healthy volunteers) were microbiologically examined. Colonization by *S. aureus*, GNB and URT pathogens were observed in 28.4%, 25.03% and 12.5% of the studied cohort, respectively.

The overall 26 isolates of *S. aureus* were cultured: 20 (80%) from nostrils and 5 (25%) from oropharynx. From 22 patients, 24 GNB isolates (two patients were colonized by two different species of GNB) were cultured: 9 (37.5%) isolates were obtained from nostrils and 15 (62.5%) from oropharynx. GNB isolates mainly belonged to the *Enterobacteriales*, namely *Enterobacter cloacae* (three isolates), *E. agglomerans* (one isolate), *Citrobacter freundi* (two isolates), *Citrobacter diversus* (two isolates), *Klebsiella pneumoniae* (three isolates), *K. oxytoca* (two isolates), *Serratia marcescens* (two isolates), *Escherichia coli* (six isolates) and only three isolates belonged to nonfermentative Gram-negative rod—*Pseudomonadaceae*: *P. aeruginosa* (two isolates) and *P. putida* (one isolate). Out of 11 patients colonized by URT pathogens, 6 patients were colonized by *Haemophilus influenzae*, 3 patients by *Streptococcus pneumoniae,* 1 patients by *Streptococcus pyogenes* and 1 patient by *Neisseria meningitidis*.

Polymicrobial colonization was significantly less common in the control group than in CLL patients (*p* = 0.0005); the proportion of CLL patients with polymicrobial colonization was 42% (21/50 patients), and 7.9% (3/38 patients) of healthy subjects were colonized by the mixture of pathogens.

The same significantly higher frequency of *S. aureus* and GNB colonization in CLL patients (40.0%) was observed in comparison to healthy volunteers (15.8% and 7.9%, respectively) (Figure 1). This difference was statistically significant for both groups (*p* = 0.018; RR 2.5; 95%CI 1.1–5.7 and *p* = 0.0006; RR 5.1; 95%CI 1.6–15.8). Natural nasal and pharyngeal microbiota were cultured from 30 healthy subjects of the control group (78.9%) and 4 CLL patients (8%) (*p* < 0.0001, RR 9.9, 95%CI 3.8–25.6).

Analysis of affecting factors for bacterial colonization in CLL patients was determined for *S. aureus*, Gram-negative rods (GNB), and upper respiratory pathogens URT pathogens, including *Haemophilus influenzae, Streptococcus pneumoniae, and Neisseria meningitidis,* separately. 

In univariate analysis, factors associated with *S. aureus* colonization were of male gender, 1^st^ Rai stage, EBV infection, duplication of lymphocytosis, higher level of B2M, as well as higher percentage of CD8^+^/PD-1^+^ cells within CD8^+^ cells (Table 4). We fitted multivariate models with all variables, EBV infection and lower frequency of CD4^+^/PD-1 were revealed as independent factors (OR 41.7, 95%CI 5.2–337.2, *p* < 0.0001 and OR 0.9, 95%CI 0.8–0.9, *p* < 0.0001).

The univariate analysis did not indicate any factors associated with GNB and URT pathogens colonization in CLL patients. However, the rate of disease progression or death within the time from the CLL diagnosis was higher in patients colonized by GNB or *S. aureus* but significant in the case of GNB colonization (Figure 2).

## 4. Discussion

In this study, we have made an attempt to complement our knowledge of the infection-prone nature of CLL. All CLL patients recruited to this study suffered mainly from URTIs, LRTIs, UTIs, and skin infections. For almost 80% of patients, two or more different types of infections were documented. Immunoglobulin deficiency is common at diagnosis and occurs spontaneously throughout the natural course of the disease; and its related disease duration and stage, moreover, is predictive of an increased frequency of infection. In CLL patients, the prevalence of hypogammaglobulinemia varies from 10 to 100% and it is related to the duration and stage of the disease. Deficiencies can be seen in all three classes of immunoglobulins -IgG, IgA and IgM [12]. Despite numerous reports correlating hypogammaglobulinemia and infection in CLL, the relationship between the level of a specific immunoglobulin class and the risk of infection is not well-established. Patients with hypogammaglobulinemia may not have infections and, on the contrary, patients with CLL and with normal Ig levels can be subject to recurrent infections; in fact, as well as low Ig levels, it is important that B lymphocytes are able to form a specific response [13,14,15]. In our study, only one-third of CLL patients had hypogammaglobulinemia in IgA (29.2%), IgG (33.85%) and IgM (27.7%), and in these groups more than 60% of patients developed LRTIs but without statistic relevance. 

There is a significant correlation between the stage of disease and infections; infectious episodes were observed not only more frequently, but also more severely in patients in stage C (82%) rather than stage A (33%) [12]. In our study, LRTIs were significantly more frequent in patients with advanced disease at diagnosis, as it was suggested by an advanced Binet stage B, Rai stage, higher lactic dehydrogenase, and beta-2-microglobulin levels.

Patients with CLL have therapy-related immune suppression from chemotherapeutics such as alkylating agents, purine analogues and monoclonal antibodies [12]. Although these drugs have dramatically improved CLL outcomes, the predisposition to serious infections can result in significant morbidity, −80% of CLL patients will have a significant infection over the course of their disease, with up to 60% of people dying from infection [15]. In our study, pneumonia and bronchitis were observed only in patients after implementation of treatment that increased the risk of infections more than 50 times.

Potential involvement of EBV in the clinical course of CLL is still unexplained. The immune response which controls latent EBV infection in healthy carriers is impaired in CLL patients and might result in poor control of reactivation and replication of the virus. Since EBV may activate B cells, stimulate their proliferation, and inhibit their apoptosis, it might be the explanation for high risk of infections in patients with EBV-DNA in PMBCs and isolated B lymphocytes in this study. This is in accordance with our previous study documented, in which increased EBV load in peripheral blood may predict the poor clinical outcome of CLL [9].

To our best knowledge, this is the first study evaluating frequencies of PD-1 positive T CD3^+^ cells (CD4^+^ and CD8^+^) and B cells (CD 19^+^) as risk factors of infections and bacterial colonization in CLL patients. All of these markers were significant risk factors of LRTIs in CLL patients. Moreover, the patients colonized with *S. aureus* had significantly higher frequencies of CD8^+^/PD-1^+^. Markers of T cell exhaustion (PD-1 and PD-L1) and T regulatory cells, potent inhibitors of T cell activation, could be used to immunophenotype the patient immune effector cells [16]. Rabe et al. demonstrated that stimulation with *S. aureus* converts neonatal CD4^+^CD25^neg^T cells into FOXP3^+^CD25^+^CD127^low^T cells and showed that APCs expressing PD-L1 were pivotal for this induction as blocking the interaction between PD-1 and PD-L1 reduced the proportion of FOXP3^+^CD25^+^CD127^low^T cells [17].

Recent interest in exploiting the PD-1:PD-L regulatory axis for treatment of chronic viral infections, cancer, and autoimmunity is supported by numerous mouse, non-human primate and human studies [18,19,20,21]. Nonetheless, remarkably little is known about how this immunoregulatory pathway influences the immune response to bacterial infections. Studies with two distinct intracellular bacteria yielded divergent results, with PD-1 suppressing protective responses to *Listeria monocytogenes* via dendritic cell regulation [22], but promoting survival in response to *Mycobacterium tuberculosis* infection via the suppression of excessive inflammation [23,24]. To date, the sole investigation into PD-1 effects on acute extracellular bacterial infection employed a cecal ligation puncture model, wherein PD-1 expression on macrophages was found to promote macrophage dysfunction and lethality due to sepsis [25]. McKay et al. [18] found that PD-1−/− mice, as well as wild type mice treated with a PD-1-blocking antibody, exhibited significantly increased survival against lethal *Streptococcus pneumoniae* infection following either priming with low-dose pneumococcal respiratory infection or *S. pneumoniae*-capsular polysaccharide immunization. Enhanced survival in mice with disrupted PD-1:PD-L interactions was explained by significantly increased proliferation, isotype switching, and IgG production by pneumococcal capsule-specific B cells. Both PD-1 ligands, B7-H1 and B7-DC, contributed to PD-1-mediated suppression of protective capsule-specific IgG. Importantly, PD-1 was induced on capsule-specific B cells and suppressed IgG production and protection against pneumococcal infection in a B cell-intrinsic manner. These results provide the first demonstration of a physiologic role for B cell-intrinsic PD-1 expression in vivo [18]. 

This study was also carried out to define the prevalence and risk factors for bacterial and fungal upper respiratory tract carriage in patients with CLL. Bacterial colonization of the respiratory tract frequently precedes the onset of serious invasive infection but also drives the evolution of opportunistic pathogens. CLL patients have been shown to have increased bacterial binding to their respiratory mucosa. Our data indicated that all of the tested bacterial colonization rates were higher in CLL patients but it was statistically significant for *S. aureus* and GNB colonization. 

The prevalence of *S. aureus* in the anterior nares of a sample of healthy Europeans at a single time point was 21.6%, with slightly higher rates in men and younger adults [26]. Certain patient populations tend to have higher rates of colonization than healthy adults. Almost all (>90%) adult patients with atopic dermatitis (AD) are *S. aureus* nares and/or skin carriers. Granulomatosis with polyangiitis (GPA—formerly Wegener’s granulomatosis) patients also have higher rates of nasal *S. aureus* carriage. Other cohorts with recurrent skin breaches have higher carriage rates, including insulin-dependent diabetics, renal replacement therapy patients, intravenous drug users, and HIV-positive patients [27]. In our study, for the first time, the prevalence of *S. aureus* colonization in CLL patients was estimated. Our results were comparable to the report of Nguyen et al. [28] where *S aureus* colonization rates in cutaneous T-cell lymphoma subjects were 44%, 48% in psoriasis subjects, and 28% in healthy control subjects. The next point of this study was to assess the correlation between S. aureus colonization and CLL disease severity. In this study, *S. aureus* colonization rates in CLL patients were affected not only by gender, but also by disease severity of the patients related to higher beta-2-microglobulin level, duplication of lymphocytosis and Rai stage. However, there were no connections between *S. aureus* colonization and the number or kind of infections in CLL patients. Curiously, GNB colonization was a bad prognostic for disease progression. Freedom from CLL diagnosis to progression was significantly different between patients GNB colonized and non-colonized.

Niederman [29] suggested that oropharyngeal colonization by GNB is a marker for critically ill patients who have multiple deficiencies in the host defense system of their respiratory tract, and these patients are also likely to have other impairments in the cellular and humoral response to bacterial invasion. This study confirmed the results of our previous one where about 8% of healthy volunteers and about 37% of CLL patients were colonized by GNB [6]. In previous study, we showed that the GNB colonization rate was higher among CLL patients with lower levels of IgG in serum and higher numbers of neutrophils or a higher number of lymphocytes in serum. The longer the time elapsed since diagnosis, the higher the frequency of GNB colonization observed [6]. In this study, no factors associated with GNB colonization were observed.

## 5. Conclusions

The data presented in this study represent the basic date for identification of further risk factors of infection and bacterial oropharyngeal colonization in CLL patients. This study highlights EBV infection and frequencies of PD-1 positive T CD3^+^ cells (CD4^+^ and CD8^+^) and B cells (CD 19^+^) as risk factors in CLL patients.

## Figures and Tables

**Figure 1 jcm-08-00861-f001:**
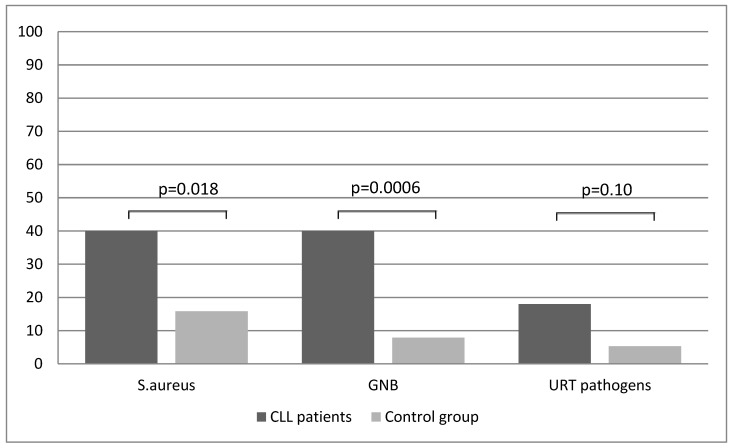
Oropharyngeal bacterial colonization in the cohort group. GNB, Gram-negative bacilli; URT, upper respiratory tract pathogens.

**Figure 2 jcm-08-00861-f002:**
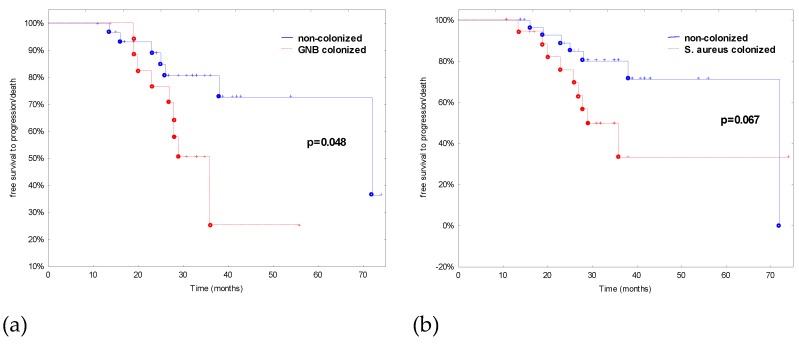
Effect of GNB (**a**) and *S. aureus* (**b**) colonization on time of free survival to disease progression/death in CLL patients.

**Table 1 jcm-08-00861-t001:** Characteristics of the cohort group.

	CLL Patients (*n* = 50)		Control (*n* = 38)		*p* Value
	Mean ± SD	Median (Range)	Mean ± SD	Median (Range)	
Age	63.1 ± 10.7	63.0 (38–80)	62.6 ± 9.9	62.5 (41–83)	0.85
Male gender, *n* (%)	19 (38.0)		15 (39.5)		1.0
No of infections per year	4.4 ± 1.7	4.0 (1–8)	1.1 ± 0.91	1.0 (0–3)	<0.0001
Type of infections					
URTIs, *n* (%)	43 (86.0)		34 (89.5)		0.75
LRTIs, *n* (%)	34 (68.0)		1 (2.6)		<0.0001
UTIs, *n* (%)	11 (22.0)		0 (0)		0.002
Skin infections, *n* (%)	10 (20.0)		0 (0)		0.004
No of antibiotic therapies (per patient)	3.7 ± 1.8	4.0 (1–8)	0.16 ± 0.37	0 (0–1)	<0.0001

**Table 2 jcm-08-00861-t002:** Clinical and laboratory parameters of the CLL patients.

	Mean ± SD	Median (Range)
URTIs (%), including:		
- Pharyngitis	40 (80.0)	
- Laryngitis	7 (14.0)	
- Otitis	19 (38.0)	
LRTIs (%), including:		
- Bronchitis	22 (44.0)	
- Pneumonia	13 (26.0)	
Genitourinary infections (%)	11 (22.0)	
Skin/soft tissue infections (bacterial/fungal) (%)	10 (20.0)	
The Rai stage, *n* (%): 0	16 (32.0)	
1	18 (36.0)	
2	16 (32.0)	
Binet stage, *n* (%): A	16 (32.0)	
B	34 (68.0)	
Splenomegaly, *n* (%)	14 (28.0)	
Hepatomegaly, *n* (%)	8 (16.0)	
Institution of treatment, *n* (%)	21 (42.0)	
Remission, *n* (%)	5 (10.0)	
Progression, *n* (%)	16 (32.0)	
EBV(+) patients, *n* (%)	25 (50.0)	
EBV EA IgA (U/mL) EBV(+)/EBV(-)	59.0 ± 125.4/6.3 ± 6.0	4.8 (2.4–580.4)/5.1 (1.2–33.3)
EBV EA IgG (U/mL) EBV(+)/EBV(-)	63.4 ± 89.0/14.0 ± 19.8	23.7 (3.9–381.1)/8.2 (2.0–94.8)
EBV EA IgM (U/mL) EBV(+)/EBV(-)	5.4 ± 4.3/4.3 ± 2.8	4.1 (1.4–19.2)/3.6 (0.8–10.5)
EBV EBNA-1 IgA (U/mL) EBV(+)/EBV(-)	12.0 ± 11.5/5.7 ± 3.6	8.3 (1.3–38.3)/5.1 (1.5–15.3)
EBV EBNA-1 IgG (U/mL) EBV(+)/EBV(-)	132.6 ± 141.8/64.8 ± 41.0	78.5 (22.5–659.3)/55.7 (23.5–181.7)
EBV EBNA-1 IgM (U/mL) EBV(+)/EBV(-)	6.9 ± 4.5/4.8 ± 2.3	6.2 (1.8–19.1)/4.8 (0.9–10.5)
EBV VCA IgA (U/mL) EBV(+)/EBV(-)	10.7 ± 9.3/7.7 ± 4.3	6.1 (1.9–34.0)/5.5 (2.3–19.8)
EBV VCA IgG (U/mL) EBV(+)/EBV(-)	203.2 ± 82.6/162.0 ± 87.8	209.2 (40.6–352.8)/149.4 (25.2–358.0)
Duplication of lymphocytosis, *n* (%)	26 (52.0)	
Leukocytosis (G/L)	41.5 ± 27.5	30.9 (12.7–128.0)
Lymphocytosis (G/L)	35.1 ± 26.3	25.5 (8.4–124.0)
Hemoglobin (g/dL)	13.4 ± 1.6	13.4 (10.0-16.6)
Platelets (G/L)	183.5 ± 55.6	177.5 (101.0–299.0)
Lactate dehydrogenase, LDH (U/L)	313.7 ± 130.8	297.0 (115.0–705.0)
Beta-2 microglobulin, B2M (mg/L)	2.7 ± 1.0	2.5 (1.1–4.9)
IgA (g/L)	1.4 ± 1.2	1.1 (0.14–5.4)
IgG (g/L)	8.7 ± 3.7	8.5 (3.2–18.6)
IgM (g/L)	0.65 ± 0.5	0.5 (0.04–2.0)
T CD3^+^cells (%)	10.8 ± 7.4	9.7 (1.2–30.6)
B CD19^+^cells (%)	84.25 ± 9.5	85.3 (51.5–97.8)
CD5^+^CD19^+^cells (%)	84.4 ± 10.3	84.2 (54.9–99.3)
CD19^+^ZAP70^+^cells (%)	16.3 ± 12.0	14.0 (0.5–47.0)
Negative (<20%)	30 (60.0)	
Positive (>20%)	20 (40.0)	
CD19^+^CD38^+^cells (%)	25.2 ± 25.3	16.4 (0.2–87.5)
Negative (<30%), *n* (%)	29 (58.0)	
Positive (>30%), *n* (%)	21 (42.0)	
ZAP70 in relation to CD38, *n* (%):		
ZAP70(-) CD38(-)	20 (40.0)	
ZAP70(+) CD38(+)	14 (28.0)	
ZAP70(+) CD38(-)	7 (14.0)	
ZAP70(-) CD38(+)	9 (18.0)	
CD19^+^CD5^+^CD23^+^(%)	79.0 ± 11.2	79.9 (54.1–95.2)
CD19^+^CD25^+^cells (%)	52.4 ± 23.1	52.3 (6.7–93.0)
CD19^+^CD69^+^cells (%)	33.5 ± 19.3	30.5 (3.9–77.6)
CD3^+^CD25^+^cells (%)	21.95 ± 13.9	20.4 (1.2–57.6)
CD3^+^CD69^+^cells (%)	4.7 ± 4.8	3.2 (0.1–21.8)
CD4^+^/PD-1^+^ (among CD4^+^) (%)	19.6 ± 9.8	16.4 (8.5–49.8)
CD8^+^/PD-1^+^(among CD8^+^) (%)	14.6 ± 7.1	14.1 (1.6–33.2)
CD19^+^/PD-1^+^(among CD19^+^) (%)	19.1 ± 7.1	19.7 (6.5–35.65)

**Table 3 jcm-08-00861-t003:** Univariate analysis of risk factors for lower respiratory tract infections (LRTIs) in CLL patients.

Factors	LRTIs (*n* = 34)	No LRTIs (*n* = 16)	OR (95%)	*p* Value
Age	63.7 ± 9.9	61.6 ± 12.4	1.01 (0.96–1.1)	0.66
Male gender:	13 (38.2)	6 (37.5)	1.03 (0.3–3.5)	1.0
The Rai stage: 0	6 (17.7)	10 (62.5)	referent	
1	14 (41.2)	4 (25.0)	5.8 (1.3–26.2)	0.022
2	14 (41.2)	5 (12.5)	11.7 (1.9–70.2)	0.007
Binet stage B	28 (82.4)	6 (37.5)	7.8 (2.0–29.8)	0.0029
Splenomegaly	12 (35.3)	2 (12.5)	3.8 (0.7–19.7)	0.087
Hepatomegaly	8 (23.5)	0 (0)	10.6 (0.6–159.9)	0.034
EBV positive	23 (67.7)	2 (12.5)	14.6 (2.8–75.9)	0.0006
*S. aureus* colonization	15 (44.1)	4 (25.0)	3.8 (0.7–19.7)	0.11
GNB colonization	15 (44.1)	4 (25.0)	0.5 (0.1–2.3)	0.23
URT pathogens	6 (17.7)	3 (18.8)	0.9 (0.2–4.3)	1.0
Duplication of lymphocytosis	23 (67.7)	3 (18.8)	9.1 (2.1–38.5)	0.002
Institution of treatment	21 (61.8)	0 (0)	52.6 (2.9–950.7)	<0.0001
Lactate dehydrogenase (U/L)	339.7 ± 137.5	258.4 ± 97.6	1.01 (1.0–1.012)	0.04
Beta-2 microglobulin (mg/L)	3.0 ± 1.0	2.03 ± 0.7	3.4 (1.4–8.0)	0.001
IgA (g/L)	1.5 ± 1.3	1.2 ± 0.99	1.3 (0.8–2.3)	0.38
IgG (g/L)	9.1 ± 3.6	7.8 ± 3.8	1.1 (0.9–1.3)	0.18
IgM (g/L)	0.6 ± 0.5	0.7 ± 0.6	0.7 (0.2–2.2)	0.72
CD19^+^ZAP70^+^cells (%)	18.4 ± 12.1	11.9 ± 10.8	1.1 (1.0–1.12)	0.098
CD19^+^CD38^+^cells (%)	30.5 ± 25.8	13.8 ± 20.7	1.03 (1.0–1.06)	0.012
CD4+/PD-1^+^ (within CD4^+^) (%)	16.1 ± 7.7	11.3 ± 4.5	1.2 (1.0–1.3)	0.001
CD8+/PD-1^+^ (within CD8^+^) (%)	21.2 ± 6.2	14.5 ± 6.7	1.1 (1.0–1.3)	0.030
CD19+/PD-1^+^ (within CD19^+^) (%)	22.2 ± 10.0	14.0 ± 6.6	1.2 (1.1–1.3)	0.001

**Table 4 jcm-08-00861-t004:** Univariate analysis of risk factors for bacterial colonization in CLL patients.

		*S. aureus*			GNB	
Factors	Colonized (*n* = 19)	Non-Colonized (*n* = 31)	OR (95%CI)	Colonized (*n* = 19)	Non-Colonized (*n* = 31)	OR (95%CI)
Age	71 (38–79)	62.0 (44–80)	1.0 (0.98–1.1)	63 (45–79)	63 (38-80)	1.0 (0.95–1.1)
Male gender:	11 (57.9%)	8 (25.8%)	3.9 (1.2–13.3) *	10 (52.6%)	9 (29.0%)	2.7 (0.8–8.9)
The Rai stage: 0	3 (15.8%)	13 (41.9%)	reference	6 (31.6%)	10 (32.3%)	reference
1	10 (52.6%)	8 (25.8%)	5.4 (1.1–25.8) *	8 (42.1%)	10 (32.3%)	1.3 (0.3–5.3)
2	6 (31.6%)	10 (32.3%)	2.6 (0.5–13.0)	5 (26.3%)	11 (35.5%)	0.7 (0.2–3.3)
Binet stage B	16 (84.2%)	18 (58.1%)	3.8 (0.9–16.0)	13 (68.4%)	21 (67.7%)	1.03 (0.3–3.5)
Splenomegaly	6 (31.6%)	8 (25.8%)	1.3 (0.4–4.7)	4 (21.1%)	10 (32.3%)	0.6 (0.1–2.1)
Hepatomegaly	4 (21.0%)	4 (12.9%)	1.8 (0.4–8.3)	1 (5.3%)	7 (22.6%)	0.2 (0.02–1.7)
EBV positive	14 (73.7%)	11 (35.5%)	5.1 (1.4–17.9) *	10 (52.6%)	15 (48.4%)	1.2 (0.4–3.7)
URTIs	18 (94.7%)	25 (80.7%)	1.5 (1.0–2.2)	17 (89.5%)	26 (83.9%)	1.2 (0.7–2.0)
LRTIs	15 (79.0%)	19 (61.3%)	1.3 (0.9–2.0)	15 (79.0%)	19 (61.3%)	1.3 (0.9–2.0)
UTIs	2 (10.5%)	8 (25.8%)	0.7 (0.5–1.1)	3 (15.8%)	7 (22.6%)	0.9 (0.5–1.4)
Skin infections	5 (26.3%)	5 (16.1%)	1.8 (0.5–7.5)	3 (15.8%)	7 (22.6%)	0.9 (0.5–1.4)
Duplication of lymphocytosis	14 (73.7%)	12 (38.7%)	4.4 (1.3–15.5) *	10 (52.6%)	16 (51.6%)	1.04 (0.3–3.3)
Institution of treatment	10 (52.6%)	11 (35.5%)	2.02 (0.6–6.5)	11 (57.9%)	10 (32.3%)	2.9 (0.9–9.4)
Lactate dehydrogenase (U/L)	346 (179–625)	245 (115–705)	1.0 (0.99–1.0)	311 (115–595)	285 (153–705)	1.0 (0.99–1.0)
Beta-2 microglobulin (mg/L)	3.1 (1–5)	2.1(1–4.5)	2.3 (1.2–4.3) *	2.3 (1–4)	2.6 (1–5)	1.2 (0.7–2.2)
IgA (g/L)	1.5 (0.1–5)	1.1 (0.2–5.4)	1.1 (0.7–1.8)	1.2 (0.2–5)	1.0 (0.1–5)	1.3 (0.8–2.2)
IgG (g/L)	8.7 (4–17)	7 (3–19)	1.05 (0.9–1.2)	8.3 (4–17)	8.7 (3–19)	1.03 (0.88–1.2)
IgM (g/L)	0.6 (0.1–2)	0.45 (0.04–2)	1.0 (0.3–3.2)	0.6 (0.1–2)	0.45 (0.04–2)	1.3 (0.4–4.2)
CD19^+^ZAP70^+^cells (%)	20.6 (2–47)	12.4 (0.5–38)	1.05 (1.0–1.1)	18.3 (0.5–47)	13.1 (0.7–45)	1.01 (0.96–1.06)
CD19^+^CD38^+^cells (%)	30.6 (0.4–87)	8.9 (0.2–5)	1.02 (1.0–1.05)	11.8 (0.2–75)	20.5 (0.4–87)	1.0 (0.98–1.02)
CD4^+^/PD-1^+^ (within CD4^+^) (%)	20.4 (9–35)	14.4 (8–50)	1.0 (0.95–1.07)	15.9 (9–50)	16.8 (8–43)	0.98 (0.92–1.04)
CD8^+^/PD-1^+^ (within CD8^+^) (%)	18.3 (2.5–33)	12.6 (1.6–29)	1.1 (1.01–1.2) *	13.2 (5–33 )	14.5 (1–29)	1.03 (0.95–1.1)
CD19^+^/PD-1^+^ (within CD19^+^) (%)	20.7 (7–30)	18.3 (6–36)	1.06 (0.97–1.2)	20.6(8–36)	19.7 (6.5–30)	1.02 (0.9–1.1)

* *p* < 0.05; URTIs, upper respiratory tract infections; LRTIs, lower respiratory tract infections; UTIs, urinary tract infections.

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
