# Peer review of "Bacterial Colonization in Patients with Chronic Lymphocytic Leukemia and Factors Associated with Infections and Colonization"

_jcm, 2019, doi:10.3390/jcm8060861_

Reviewer 1 Report

Summary:

The authors present a descriptive analysis of microbial spectra observed in CLL patients compared to controls. Colonization of oral cavity was analyzed by conventional microbiology and compared to a multitude of clinical and immune parameters. Associations between multiple immune parameters and S. aureus and gram neg bacilli (GNB) as well as with disease progression are reported. An association with EBV replication was highlighted.

General comments:

Overall, the topic is of clear interest, since microbiota seem to be important determinants of immune interaction and thus may be of importance in CLL, with its multitude of immune interactions. As such the current manuscript has merits in thoroughly describing a cohort. It, however, suffers from a couple of problems that need to be addressed.

Major criticism:

The 2 major problems are:

While the CLL cohort has a relevant size, the control population is much too small to capture the relevant heterogeneity of the microbiota composition in a population. There are many factors that have been reported to affect the rates of S. Aureus colonization or GMB colonization reported in the literature. It is unlikely that the 15 controls can really be representative of a normal population. Any bias in the selection of such a small group will then produce positive results in statistical tests (especially if many are performed (see below)).

The authors have meticulously compiled a large dataset (which they can be commended for) and proceed to analyze it in a multitude of statistical comparisons (over 50 p-values from tables 1 and 3 and 50 comparisons in Table 4). It does not seem, that the authors have employed any correction for multiple testing – or other methods of reducing the rate of accidental association (multivariate tests).

It is thus clear that any discussion of these data will have a high probability of following chance findings. This needs to be addressed before the manuscript can be a useful addition to the literature.

Since some of the levels of significance seem high, it is likely that some parameters may be carried into significance even after a multiple testing correction (even a bonferroni correction may be sufficient). The resulting data would then be easier to discuss with more confidence. Also, importantly, the presented parameters are unlikely to be independent (a lot of them may be indirect surrogates for tumor mass) – they should thus not be treated as such and a more complex model (e.g. multivariate models)

Overall, some consultancy with a statistician should be extremely helpful to improve the paper.

In addition, and ideally after increasing the number of controls, the control population needs to be described in more detail (how were they selected, how representative are they of the general population and the CLL cohort presented in their composition). Specifically the difference in type of infections will be very affected by the composition and size of the cohort (e.g. the larger percentage of men in the control, will result in lower urinary tract infections, without any other influence)

Regarding the EBV question, it may be important to reference the number of EBV positive patients from serology, since replication (even at low levels) may be more of a marker for immune-deficiency states or active infections (with reactive reactivation of EBV) than a correlation with EBV itself. If the data cannot be obtained this needs to be discussed.

How were the infections reported – is this a retrospective analysis chart reviews ore even pure anamnesis may have limited quality for detecting infections.

The effects on progression need to be looked at in a multivariate model, to be valid for reporting, since there are so many covariates that affect (or are correlated with) this outcome.

Finally, the discussion is much too long and should be shortened significantly.

Minor (but annoying) points:

A discussion of the role of hypogammaglobulinemia in colonization rates should be included, based on literature or own data.

I am not sure for which of the variables in this patient analyses a mean value would ever be the relevant and correct parameter to report. Medians would certainly be more appropriate for most parameters (see also the statistician comment above).

Table 1 states a “no of antibiotic therapies” is this per patient, per cohort, per day of observation – this is not clarified.

Also, the table states that 40% of patients needed treatment at some point – were all infections exclusively before treatment, or might there be an effect of treatment?

Some patients have multiple samples – is this equally distributed between CLLs and controls – otherwise this will create significant bias.

The staph aureus colonization rate in Figure 1 seems very low in the control group – is there a benchmark from Poland, that would suggest that this does not point to a bias?

Author Response

We would like to thank you for your suggestions of revisions, which undoubtedly improved our paper. We have revised the manuscript in accordance with your comments hoping that in current form will be acceptable for publication.

While the CLL cohort has a relevant size, the control population is much too small to capture the relevant heterogeneity of the microbiota composition in a population. There are many factors that have been reported to affect the rates of S. Aureus colonization or GMB colonization reported in the literature. It is unlikely that the 15 controls can really be representative of a normal population. Any bias in the selection of such a small group will then produce positive results in statistical tests (especially if many are performed (see below)).

We extended the control group to 38 healthy volunteers. All calculations considering the comparison of bacterial colonization in CLL patients and healthy volunteers were updated and showed no changes in the final conclusions.

 Given the fact that risk factors of bacterial colonization were determined for CLL patients only we decided to remove the comparison of laboratory parameters in control group which was presented in Table 1. We reconstructed the Table 1 and Table 2.

The authors have meticulously compiled a large dataset (which they can be commended for) and proceed to analyze it in a multitude of statistical comparisons (over 50 p-values from tables 1 and 3 and 50 comparisons in Table 4). It does not seem, that the authors have employed any correction for multiple testing – or other methods of reducing the rate of accidental association (multivariate tests).

It is thus clear that any discussion of these data will have a high probability of following chance findings. This needs to be addressed before the manuscript can be a useful addition to the literature.

Since some of the levels of significance seem high, it is likely that some parameters may be carried into significance even after a multiple testing correction (even a bonferroni correction may be sufficient). The resulting data would then be easier to discuss with more confidence. Also, importantly, the presented parameters are unlikely to be independent (a lot of them may be indirect surrogates for tumor mass) – they should thus not be treated as such and a more complex model (e.g. multivariate models)

Overall, some consultancy with a statistician should be extremely helpful to improve the paper.

We have already provided multivariate analysis (logistic regression) looking for the risk factors associated with bacterial colonization but no regression model was built.

In addition, and ideally after increasing the number of controls, the control population needs to be described in more detail (how were they selected, how representative are they of the general population and the CLL cohort presented in their composition). Specifically the difference in type of infections will be very affected by the composition and size of the cohort (e.g. the larger percentage of men in the control, will result in lower urinary tract infections, without any other influence)

We matched all the patients in CLL group with 38 of healthy volunteers according to sex and age (Table 1). The study group consisted of healthy people who came to the outpatient clinic to perform periodic examinations, which are necessary to obtain a work permit. After excluding the existence of autoimmune diseases, allergies, cancers, and chronic infections, these people were included in the study as a control group. None of the controls used immunomodulating agents or hormonal preparations, showed signs of infection within at least 3 months prior to the study, underwent blood transfusion, had a history of oncological therapy or prior treatment for tuberculosis or other chronic conditions that could be associated with impaired cellular or humoral immunity. They were unvaccinated against Streptococcus pneumoniae, Haemophilus influenzae, Neisseria meningitidis or seasonal influenza. We received written constent from them to participate in the study and a declaration that they would report every 3-4 months or if an infection occurred.

Regarding the EBV question, it may be important to reference the number of EBV positive patients from serology, since replication (even at low levels) may be more of a marker for immune-deficiency states or active infections (with reactive reactivation of EBV) than a correlation with EBV itself. If the data cannot be obtained this needs to be discussed.

We added the values of Ig levels for EBV positive and negative patients separately.

How were the infections reported – is this a retrospective analysis chart reviews ore even pure anamnesis may have limited quality for detecting infections.

Number of infections were reported from case report forms. The patients were observed from the time of CLL diagnosis. Observation period amounted to 26 months (min. 12.50 – max. 60), mean 28.25±16.44. All persons included in the study and control group were observed in the Immunological Clinic by a specialist in the field of clinical immunology every 3-4 months or more often - in the case of an infection. On this basis, we obtained information on the frequency and type of infections.

The effects on progression need to be looked at in a multivariate model, to be valid for reporting, since there are so many covariates that affect (or are correlated with) this outcome.

We provided Cox hazard regression in order to find covariates that affect disease progression/death in CLL patients. However no variables were selected in the model building procedure.

Finally, the discussion is much too long and should be shortened significantly.

 The discussion was shortened together with references.

Minor (but annoying) points:

A discussion of the role of hypogammaglobulinemia in colonization rates should be included, based on literature or own data.

Hypogammaglobulinemia is the most common disorder in CLL (Molica S. Infections in chronic lymphocytic leukemia: risk factors, and impact on survival, and treatment. Leuk Lymphoma. Apr 1994, vol. 13(3-4), pp. 203-14). It occurs with a variable frequency in 10 to 100% of patients and the percentage of patients with this disorder increases with the duration of the disease, its severity and the use of immunochemotherapy, moreover, hypogammaglobulinemia does not disappear even when complete remission is achieved, although recently there have been reports of possible partial reconstitution of immunoglobulin production in patients treated with high doses of rituximab (Nosari A. Infectious complications in chronic lymphocytic leukemia. Mediterr J Hematol Infect Dis. 2012, vol. 4(1), pp. e2012070) (Morrison VA. Infectious complications of chronic lymphocytic leukemia: pathogenesis, spectrum of infection, preventive approaches. Best Pract Res Clin Haematol. Mar 2010, vol. 23(1), pp. 145-53). In the course of CLL, there is an increase in the number of monoclonal B lymphocytes that show anergy (PD-1+ B cells), and the number of normal B lymphocytes is significantly reduced (Molica S. Infections in chronic lymphocytic leukemia: risk factors, and impact on survival, and treatment. Leuk Lymphoma. Apr 1994, vol. 13(3-4), pp. 203-14). In addition, there is a disturbance within the T-cell subpopulation performing an auxiliary function for B-lymphocytes. An abnormal B-cell response to IL-2 is also found. The most common in CLL patients is the reduction of all immunoglobulin classes IgA, IgG, IgM, and within the IgG subclasses, the deficiency most often affects IgG3 and IgG4 (Crassini KR, Zhang E, Balendran S, Freeman JA, Best OG, Forsyth CJ, Mackinlay NJ, Han P, Stevenson WS, Mulligan SP. Humoral immune failure defined by immunoglobulin class and immunoglobulin G subclass deficiency is associated with shorter treatment-free and overall survival in Chronic Lymphocytic Leukaemia. Br J Haematol. 2018 Apr;181(1):97-101. doi: 10.1111/bjh.15146.). Patients with CLL with hypogammaglobulinemia usually get recurrent bacterial infections similar to those with primary hypogammaglobulinemia. Despite many reports on dependencies between hypogammaglobulinemia and infections, this problem is not well understood. In some patients with immunoglobulin deficiency, there is no increased susceptibility to infection, and in some patients with normal immunoglobulin levels they often occur. This suggests that the ability of the normal (non-leukemic) B cell populations to play a specific immune response and the production of specific antibodies plays a large role. As a rule, this ability is impaired in all patients (Pasiarski M, Rolinski J, Grywalska E, Stelmach-Goldys A, Korona-Glowniak I, Gozdz S, Hus I, Malm A. Antibody and plasmablast response to 13-valent pneumococcal conjugate vaccine in chronic lymphocytic leukemia patients--preliminary report. PLoS One. 2014 Dec 15;9(12):e114966. doi: 10.1371/journal.pone.0114966. eCollection 2014). Recurrent infections caused by Streptococcus and Haemophilus are particularly strongly associated with a deficiency of IgG (Nosari A. Infectious complications in chronic lymphocytic leukemia. Mediterr J Hematol Infect Dis. 2012, vol. 4(1), pp. e2012070) immunoglobulins. The most significant relationship between IgG level and infections occurs in people whose IgG levels fall below 300 mg / dl (Boughton BJ, Jackson N, Lim S, Smith N. Randomized trial of inravenous immunoglobulin prophylaxis for patients with chronic lymphocytic leukaemia and secondary hypogammaglobulinaemia. Clin Lab Haematol. Mar 1995, vol. 17(1), pp. 75-80).

I am not sure for which of the variables in this patient analyses a mean value would ever be the relevant and correct parameter to report. Medians would certainly be more appropriate for most parameters (see also the statistician comment above).

It is true that most of variables had not normal distribution but we decided to show both parameters for detailed analysis.

Table 1 states a “no of antibiotic therapies” is this per patient, per cohort, per day of observation – this is not clarified.

Number of antibiotic therapies were presented per patients.

Also, the table states that 40% of patients needed treatment at some point – were all infections exclusively before treatment, or might there be an effect of treatment?

Our study included 50 newly diagnosed, previously untreated patients with CLL. It was a prospective study. Observation period has 26 months (min. 12.50 – max. 60), mean 28.25±16.44.

Some patients have multiple samples – is this equally distributed between CLLs and controls – otherwise this will create significant bias.

Three samples (throat and nasal swabs, and blood sample) were taken from all CLL patients and healthy volunteers.

The staph aureus colonization rate in Figure 1 seems very low in the control group – is there a benchmark from Poland, that would suggest that this does not point to a bias?

After the enlarged of the control group the rate of S.  aureus colonization meet the referred  count in  the anterior nares of 20–80% of the human population.

Reviewer 2 Report

This is a potentially interesting manuscript on colonization in patients with CLL.

There has been little done in this area and the field is well developed in other areas.

The work seems quite well done

 - the Control group is very small at n=15. This should be extended. Ideally to a similar level of the CLL group

- the association with EBV load if of interest but appears to come from nowhere. It is not mentioned in the abstract until the summary. What was the rationale for doing this ?

- how did the carriage rates correlate with antibiotic usage

- is there an assessment of sensitivity of assessment of carriage ?

- how many samples were taken per patient - were they are all at entry onto the scheme ?

- it was not clear when the vaccinations were done - initially it states that no patients were vaccinated.

Author Response

We would like to thank you for your suggestions of revisions, which undoubtedly improved our paper. We have revised the manuscript in accordance with your comments hoping that in current form will be acceptable for publication.

Please find below detailed responses to each of the comments:

This is a potentially interesting manuscript on colonization in patients with CLL.

There has been little done in this area and the field is well developed in other areas.

The work seems quite well done

 - the Control group is very small at n=15. This should be extended. Ideally to a similar level of the CLL group

We extended the control group to 38 healthy volunteers. All calculations considering the comparison of bacterial colonization in CLL patients and healthy volunteers were updated.

- the association with EBV load if of interest but appears to come from nowhere. It is not mentioned in the abstract until the summary. What was the rationale for doing this ?

Thank you very much for this suggestion. We have added the appropriate sentence, i.e. “Information regarding baseline characteristics, the Rai staging system, hematological tests results, immunophenotype of basic lymphocyte subsets, including the expression of programmed cell death-1 protein (PD-1) and its ligand (PD-L1), as well as Epstein-Barr virus (EBV) status were determined to analyze risk factors for infections and bacterial colonization”.

- how did the carriage rates correlate with antibiotic usage

There was statistically significance of antibiotic therapy neither for S. aureus nor GNB colonization (p=0.11 and p=0.19, respectively).

- is there an assessment of sensitivity of assessment of carriage ?

In our study we determined the bacterial colonization only. The swab samples were taken fro the patients only once so we are not able to assess a carriage of tested bacteria.

- how many samples were taken per patient - were they are all at entry onto the scheme ?

Three samples (throat and nasal swabs, and blood sample) were taken from all CLL patients and healthy volunteers.

- it was not clear when the vaccinations were done - initially it states that no patients were vaccinated.

Thank you for your insightful review. All of the study participants were unvaccinated. Due to preparing an another paper considering the influence of vaccination to bacterial colonization in CLL patients we mistakenly included the results to one of the tables. We deleted all data connected with the vaccination.

Round  2

Reviewer 2 Report

The authors have made a large number of improvements and the paper is now much better. Indeed it is a valuable contribution to the literature on CLL. 

I would advise reducing the numbers to fewer significant figures.